# Nanoencapsulation of buriti oil (*Mauritia flexuosa L.f.*) in porcine gelatin enhances the antioxidant potential and improves the effect on the antibiotic activity modulation

**Neyna de Santos Morais**[1], **Thaís Souza Passos**[2], **Gabriela Rocha Ramos**[3], **Victoria Azevedo Freire Ferreira**[3], **Susana Margarida Gomes Moreira**[4], **Gildácio Pereira Chaves Filho**[4], **Ana Paula Gomes Barreto**[3], **Pedro Ivo Palacio Leite**[5], **Ray Silva de Almeida**[5], **Cícera Laura Roque Paulo**[5], **Rafael Fernandes**[6], **Sebastião Ânderson Dantas da Silva**[1], **Sara Sayonara da Cruz Nascimento**[7], **Francisco Canindé de Sousa Júnior**[1,3], **Cristiane Fernandes de Assis**[1,3] *

1 Nutrition Postgraduate Program, Center for Health Sciences, Federal University of Rio Grande do Norte, Natal, RN, Brazil, 2 Department of Nutrition, Center for Health Sciences, Federal University of Rio Grande do Norte, Natal, RN, Brazil, 3 Department of Pharmacy, Center for Health Sciences, Federal University of Rio Grande do Norte, Natal, RN, Brazil, 4 Department of Cellular and Molecular Biology, Biosciences Center, Federal University of Rio Grande do Norte, Natal, RN, Brazil, 5 Department of Chemical Biology, Regional University of Cariri, Crato, CE, Brasil, 6 Chemical Intitute, Federal University of Rio Grande do Norte, Natal, RN, Brazil, 7 Postgraduate Program in Biotechnology—RENORBIO, Biosciences Center, Federal University of Rio Grande do Norte, Natal, RN, Brazil

\* cristianeassis@hotmail.com

## Abstract

The present study evaluated the cytotoxicity, antioxidant potential, and antimicrobial effect on the antibiotic activity modulation of gelatin nanoparticles containing buriti oil (OPG). The cytotoxicity analysis was performed on Chinese Hamster Ovary Cells (CHO) using a MTT [3-(4,5-dimethylthiazol-2-yl)-2,5-diphenyltetrazolium bromide] test. The antioxidant potential of buriti oil and OPG was determined by total antioxidant capacity, reducing power, and the ABTS (*2,2'-azinobis-3-ethylbenzothiazoline-6-sulfonic acid*) test. The modulating antimicrobial activity was evaluated by determining the minimum inhibitory concentration (MIC) concentration against *Escherichia coli*, *Pseudomonas aeruginosa*, *Staphylococcus aureus*, gentamicin and norflaxacillin. The nanoformulation of OPG did not show a cytotoxic effect on CHO cells and had a higher antioxidant potential than free buriti oil (p<0.05). The combination of antibiotics with free buriti oil and OPG was more efficient in inhibiting *E. coli* and *P. aeruginosa* than isolated norfloxacillin and gentamicin (p<0.05). Regarding the inhibition of *S. aureus*, OPG in combination with norfloxacillin reduced MIC by 50%. Nanoencapsulation was a viable alternative to enhance functionality and adding commercial value to buriti oil.

## Introduction

Mauritia palm or buriti (*Mauritia flexuosa L.f.*) is an abundant palm tree in South America, predominant in extensive areas of Brazil and native to areas of the Amazon Forest and Cerrado

**Data Availability Statement:** All relevant data are within the paper and its Supporting Information files.

**Funding:** This study was partly financed by the Coordenação de Aperfeiçoamento de Pessoal de Nível Superior - Brasil (CAPES) - Finance Code 001. The authors want to thank by Plantus LTDA (Nísia Floresta, Brazil) for providing the quinoa oil utilized to conduct this research. There was no additional external funding received for this study.

**Competing interests:** The authors have declared that no competing interests exist.

biome [1]. Buriti fruit is considered as a functional food due to its high levels of carotenoids. The pulp is regarded as one of the primary sources of carotenoids found in Brazilian flora diversity, with emphasis on β carotene, which has provitamin activity A [2]. In addition to containing minerals (calcium, zinc, sodium), fibers, proteins, and unsaturated fatty acids [3].

Buriti oil can be obtained from the fruit by extracting the pulp and/or pulp and peel by conventional methods (mechanical and chemical extraction) [4–6], capable of influencing the product characteristics. It is of great interest in the cosmetic and food product industries because it has a wide and varied application due to its chemical composition [7].

The oil contains vitamin E considered an important antioxidant capable of interrupting the action of free radicals and protecting the cytoplasmic membranes from oxidation, reducing pre-cancerous lesions [8–10]. In addition, it also has unsaturated fatty acids, which are associated with reduced levels of triglycerides and total cholesterol, consequently constituting a protective factor for cardiovascular diseases [11] and a protective effect on platelet activation and thrombosis [12]. It also has ascorbic acid (vitamin C), which helps form bone and teeth and acts on the immune system, preventing flu, diabetes, and scurvy [13].

Furthermore, vegetable oils have a potential effect against microorganisms such as bacteria [5]. Thus, the search for new compounds with antibacterial activity has been the objective of some research to solve the problem associated with bacterial resistance, which has become common in recent decades, increasing the concern of health authorities [14,15]. In this context, natural products have been an essential tool for being a source of new compounds with antibacterial properties, which can modulate and increase the activity and efficiency of conventional antibiotics or be resistance modifying agents [15].

Thus, it is well-known that buriti oil has numerous biological properties [16]. However, bioactive compounds are susceptible to degradation due to sensitivity to processing and storage factors such as temperature, pH, presence of light, and oxygen [17,18]. The oil's oxidation process causes unpleasant tastes, loses nutrients, and produces many toxic compounds (hydroperoxides and aldehydes) that cause mutation, aging, stroke, emphysema, heart disease, and cancer [19]. Another technological obstacle associated with the use of buriti oil in foods is associated with its lipophilic nature, which makes it challenging to apply in products with aqueous matrices, limiting the development of new products [20].

In this context, nanoencapsulation is a strategy that can be used to protect substances against environmental factors (heat, light, oxygen, humidity), improve stability, acceptability, and handling conditions, restrict contact with other components and control the release and delivery of bioactives to the target site. In addition, reducing the size to the nanometric scale ($< 100$ nm) facilitates dispersion of lipophilic substances in an aqueous matrix due to the increase in the contact surface. It can also preserve or enhance the biological properties of bioactive compounds [19–22]. Thus, nanoencapsulation can increase the antioxidant and antimicrobial potential, among other functionalities, reducing the concentration of the bioactive of interest necessary to achieve the desired effect [23].

Considering the properties present in vegetable oils, especially buriti oil, and the current needs of the food and pharmaceutical industry, our research group has been developing studies with nanoencapsulated vegetable oils [20,22]. Castro et al. [20] investigated the nanoencapsulation of buriti oil using different encapsulating agents, namely porcine gelatin (OPG) and the combination of sodium alginate and porcine gelatin (OAG), to evaluate the effect on water dispersibility and antimicrobial activity. The results showed that OPG presented the smallest particle size [51.0 (6.07) nm] and the best chemical interaction between the materials present in the system. As a result, OPG showed greater dispersibility in water [85.62% (7.82)] and potentiated the antimicrobial activity of buriti oil in 59%, 62%, and 43% against the bacteria *Pseudomonas aeruginosa*, *Klebsiella pneumonia*, and *Staphylococcus aureus*, respectively.

Therefore, due to the potential of buriti oil both from an economic and biotechnological point of view and the results obtained by Castro et al. [20], it is of fundamental importance to deepen the knowledge regarding the effects of nanoencapsulation on the biological properties of buriti oil. The resistance of microorganisms to antibiotics has increased interest in research that seeks new drugs with antimicrobial potential or that act by modulating the activity of existing antibiotics. This study aimed to assess the cytotoxicity, antioxidant potential, and modulating action on antibiotics of nanoparticles based on porcine gelatin containing buriti oil.

## Materials and methods

### Materials

Buriti oil from the *Mauritia flexuosa* species was supplied by the Plantus® S.A. cosmetics company (Nísia Floresta, in the Rio Grande do Norte, Brazil) with an IBD certificate (Agricultural and Food Inspections and Certifications). The samples were transported under protection from light and refrigerated to the Bromatology Laboratory of the Department of Pharmacy at the Federal University of Rio Grande do Norte (UFRN), where they were stored at 4˚C.

Porcine gelatin (Type A), the surfactant Tween 20, 2,2'-azinobis (3-ethylbenzothiazoline-6-sulfonic acid) (ABTS), and resazurin obtained from Sigma-Aldrich®.

### Microorganisms

Multiresistant strains of *Staphylococcus aureus* 10, *Pseudomonas aeruginosa* 24, and *Escherichia coli* 06 were supplied by the Laboratory of Microbiology and Molecular Biology of the Regional University of Cariri—URCA (Crato—CE, Brazil). The Chinese hamster ovary cell line (CHO-K1 cells) was purchased from ATCC® CCL-61™ (American Type Culture Collection), and Dulbecco's Modified Eagle Culture Medium (DMEM; Gibco, Gaithersbug, MD, USA) was used, while 3–4,5-Dimethyl-thiazol-2-yl-2,5-diphenyltetrazolium bromide (MTT) was purchased from Invitrogen (MTT; Invitrogen, Oregon, USA).

### Methods

#### Obtaining the nanoformulation containing buriti oil

The nanoformulation based on porcine gelatine containing buriti oil (OPG) was obtained through the technique of O/W emulsification followed by dispersion of a solution containing encapsulating agent in the obtained emulsion based on Medeiros et al. [24] with modifications proposed by Castro et al. [20]. Porcine gelatin (Sigma®) and Tween 20 (Sigma®) were used as encapsulating agent and surfactant, respectively. After the encapsulation process, the resulting emulsion was subjected to drying by lyophilization (LioTop L101) at -57˚C and pressure of 43 µHg.

#### Characterization of the obtained nanoparticles

The new batch of OPG obtained for this study was characterized to determine its morphology, particle diameter, and chemical interactions by scanning electron microscopy, Dynamic Light Scattering, and Fourier Transform Infrared Spectroscopy, respectively, according to Castro et al. [20]. Furthermore, it was also evaluated concerning the distribution of charges on the surface of the particles and in terms of thermal resistance by Zeta Potential, Thermogravimetry, and Differential Thermal Analysis, respectively.

### Scanning electron microscopy (SEM)

OPG was suspended in acetone, and the dispersion was dropped onto silicon plates attached to stubs using carbon tape. It was subsequently analyzed at different magnifications in high vacuum at 2–3 kV and without metallization using a FEG-SEM ZEISS microscope (AURIGA).

### Dynamic light scattering (DLS)

The nanoformulation was subjected to crosslinking with glutaraldehyde to measure the diameter of the particles, according to Castro et al. [20], to promote particle deagglomeration. It was subsequently dispersed using an ultrasonic bath in DMSO (dimethylsulfoxide). Measurements were performed in triplicate (60 s each) using only 2 mL of suspension. Data were analyzed using the NANO-flex Control 0.9.7 software program. The entire experiment was carried out in triplicate. The data obtained referring to the mean and standard deviation were plotted in the Origin® 8 program to obtain the histogram with the particle size distribution.

### Fourier transform infrared spectroscopy (FTIR)

Buriti oil, porcine gelatin, Tween 20, and OPG were homogenized separately with potassium bromide (KBr). Then they were macerated and pressed to obtain pellets. The spectra were recorded in transmittance with the mid-infrared region (400 to 4000 cm$^{-1}$). A Shimadzu spectrometer (FTIR-8400S, IRAFFINITY-1 series, IR SOLUTION version 1.60 software) was used with a scan number of 32 and resolution of 4 cm$^{-1}$.

### Zeta potential

The Zeta Potential measurements of the OPG nanoformulation were determined using a STABINO II Particle charge Titration device (Colloid Metrix). First, 10 mg of the sample was diluted in 10 mL of ultrapure water ($\geq$ 18 M$\Omega$ cm$^{-1}$) and then transferred to the cylindrical Teflon cell. The zeta potential measurement with pH variation (similar to a titration) consisted of individually adding aliquots (10 µL) of a strong acid (HCl—0.1 M) or a strong base (NaOH–0.025 M).

### Thermogravimetry (tg) and differential thermal analysis (DTA)

We employed thermoanalytical analyzes to evaluate the buriti oil, Tween 20, porcine gelatin, and OPG nanoformulation. Thus, from 6 to 7 mg of each material were evaluated in a Shimadzu® DTG-60 thermal analyzer with a heating rate of 10˚C.min$^{-1}$ in the temperature range of 30˚C to 800˚C in a nitrogen atmosphere with a flow of 50 mL.min$^{-1}$.

It is noteworthy that the variations in the residual mass of the materials concerning the variation in time and temperature were provided by the equipment. The Origin Pro Graphing & Analysis version 9.8.0.200 software program from OriginLab® was used to build the TG and DTA graphs.

### Incorporation efficiency (%)

The incorporation efficiency was determined based on El-Messery et al. [25] with modifications. 15 mL of hexane in 1.5 g of the nanoformulation were added under agitation in a TE-421 rotary incubator (Tecnal®) for 2 min, and then the mixture was filtered on Whatman No.1 filter paper.

Next, the content retained on the filter paper was washed twice with 20 mL of hexane. The liquid phase was collected, and the hexane was removed in an incubator at 60˚C for 48 h. The remaining extract represents the unencapsulated oil present on the surface of the encapsulates.

The amount of nanoencapsulated oil was obtained through the difference between the initial amount of oil used to promote the nanoencapsulation process and the free oil present on the surface of the nanoparticles after the hexane evaporates, as described below (Eq 1). The average amount of OPG obtained from the triplicate produced and measured with the aid of an analytical balance was considered (Edutec, EEQ9003F-B).

$$EO\ (g) = OUE - RO \qquad\qquad \text{Eq (1)}$$

EO: Quantity of encapsulated oil; OUE: Weight of oil used in encapsulation; RO: Weight of remaining oil.

The encapsulation incorporation was calculated using Eq 2, described below according to El-Messery et al. [25], considering the average amount of OPG obtained from the triplicate produced and measured using an analytical scale (Edutec, EEQ9003F-B).

$$EI\ (\%) = \left[\frac{EO}{IO}\right] x100 \qquad\qquad \text{Eq (2)}$$

EI: Encapsulation incorporation; EO: Amount of encapsulated oil; IO: Initial oil quantity.

## Cytotoxicity evaluation

**MTT assay.** Chinese hamster ovary (CHO) cells were seeded in 96-well plates at a density of $2x10^3$ cells.$mL^{-1}$ and cultured in α-MEM medium supplemented with 10% FCS (fetal bovine serum), 1% antibiotic and antimycotic solution, and 1% glutamine (basal medium).

At the end of 24 hours, the medium was changed with the medium containing the buriti oil and OPG samples at concentrations 50, 100, and 500 μg.$mL^{-1}$, and cytotoxicity was evaluated after 24, 48, and 72 hours. Cells maintained in the basal medium were used as a negative control. At the end of each time, 1 mg.$mL^{-1}$ of MTT solubilized in PBS buffer was added, the supernatant was aspirated after incubation at 37°C for 4 hours, and the formazan crystals solubilized with 200 μl of DMSO. Absorbance was measured at 570 nm in a plate reader (BioTek, μQuant model), and cell viability was calculated according to Eq 3 described below.

$$MTT\ reduction(\%) = \frac{Abs\ treatment}{Abs\ negative\ control} \ x\ 100 \qquad\qquad \text{Eq (3)}$$

## Antioxidant activity

**Total antioxidant capacity (TAC).** The method proposed by Prieto et al [26]. was used to determine TAC (Total antioxidant capacity). In each test tube, 100 μL of 40 mM sulfuric acid-ammonium molybdate, 280 mM sodium phosphate, and 100 μL of the sample solutions were added at a concentration of 2 mg.$mL^{-1}$ and then 700 μL of distilled water in the test tubes.

Then, the tubes were shaken and incubated in a water bath (QUIMIS, Mod. Q334M-28) at 90°C for 90 minutes to subsequently take the absorbance reading by spectrophotometry (Biospectrum, Mod. SP-220) at 695 nm. The experiment was carried out in triplicate using solutions containing the encapsulating agents as a control. Antioxidant activity was expressed in milligrams of ascorbic acid per gram of sample (mg AA.g of sample$^{-1}$). The standard curve was constructed using different ascorbic acid concentrations (25–250 mg.$g^{-1}$).

**Reducing power test.** The reducing power of the samples was quantified as described by Wang et al. [27]. First, 4 mL of a reaction mixture containing different buriti oil sample (1–50 mg.$mL^{-1}$) and OPG (3–20 mg.$mL^{-1}$) concentrations in phosphate buffer (0.2 M, pH 6.6) were incubated with potassium ferrocyanide (1% m/v) at 50°C for 20 minutes. The reaction was terminated by a TCA solution (10% m/v). The solution was then mixed with distilled water and iron chloride (0.1% m/v), and the absorbance was measured at 700 nm. The control was

carried out with the encapsulating agent. The result was expressed in the reduction of $Fe^{+3}$ to $Fe^{+2}$ (%).

**Antioxidant activity by 2,2'-azinobis-3-ethylbenzothiazoline-6-sulfonic acid (ABTS•).**
The ability to scavenge the 2,2'-azinobis-3-ethylbenzothiazoline-6-sulfonic acid (ABTS•) radical cation was determined by adapting the methodology described by Rufino et al. [28]. The ABTS• solution was prepared by adding ABTS• (7 mM) and potassium persulfate (2.45 mM) and incubated at room temperature in the dark for 16 h. Next, the ABTS solution (1 mL) was diluted in DMSO to obtain an absorbance of 0.8, at a wavelength of 734 nm [29,30]. Finally, 200 μL of this ABTS• radical solution was carefully added to the microplate well together with 40 μL of the evaluated samples.

The control was prepared to contain 200 μL of ABTS and 40 μL of DMSO. The absorbance reading at 734 nm was performed in a microplate reader (Loccus–Model LMR FLEX). After reading the absorbances, the antioxidant activity (AA) percentage was estimated using Eq 4 described below.

$$ABTS \; radical \; inhibition = 100 \; X \; \frac{Abs \; control - Abs \; sample}{Abs \; control} \qquad \text{Eq (4)}$$

The linear regression curve was determined to calculate the concentration required for the samples (crude oil and OPG) to inhibit 50% of the ABTS radical (IC 50).

## Determination of the modulating antimicrobial activity of antibiotics

The antimicrobial activity test was performed by determining the Minimum Inhibitory Concentration (MIC) of antibiotics (norfloxacin and gentamycin) and modulating activity using buriti oil and OPG (dilution of in DMSO, in the concentration of 1024 μg.mL$^{-1}$).

The bacteria used were multiresistant *Pseudomonas auriginosa* 24, *Staphylococcus aureus* 10, and *Escherichia coli* 06 strains collected in exams provided by the Regional University of Cariri (Juazeiro do Norte/CE). The origin and resistance profile of these strains were described by Bezerra et al. [31].

Bacteria cultures of $10^7$ CFU.mL$^{-1}$ kept in agar were subcultured in brain-heart broth (Brain Heart Infusion—BHI) at 37°C for 24 hours. Then, the distribution medium was serially prepared at 1:10, and 100 μL were added into Eppendorfs of BHI medium in 900 μL of inoculum. This distribution medium was transferred to the microplate, 100 μL in each well. Norfloxacin and gentamycin antibiotics were added to the test, and their inhibitory effects were investigated in their combinations with buriti oil and OPG. Both concentrations of antibiotics, crude buriti oil and OPG, were fixed at 1024 μg.mL$^{-1}$ with a dilution of 100 μL in octuplicate. Controls were included in the trials, namely the medium and the inoculum, the porcine gelatin, and the Triton X-100. The filled plates were incubated at 35 (±2)°C for 24 hours [32].

Next, an indicator solution of sodium resazurin was prepared in sterile distilled water at a concentration of 0.01% (m/v) to evidence the MIC of the samples. After incubation, 20 μL of the indicator solution was added to each well and the plates underwent an incubation period of 1 hour at room temperature. A change from blue to pink color due to the reduction of resazurin indicated bacterial growth at 37°C [32]. MIC was defined as the lowest concentration capable of inhibiting microbial growth, as evidenced by the unaltered blue color.

According to the Eqs 5, 6 and 7 below, the fractional inhibitory concentration index (FICi) measurement was determined to demonstrate the synergistic effect. The FICi index was interpreted according to the European Committee for Antimicrobial Susceptibility Testing (30),

considering the synergistic effect FICi $\leq$ 0.5 [33].

$$FIC\ buriti\ oil\ or\ OPG = \frac{MIC\ of\ buriti\ oil\ or\ OPG\ in\ combination}{MIC\ of\ buriti\ oil} \qquad \text{Eq (5)}$$

$$FIC\ antibiotic = \frac{MIC\ of\ antibiotic\ in\ combination}{MIC\ of\ antibiotic} \qquad \text{Eq (6)}$$

$$FIC\ index\ (FICi) = (FIC\ oil\ buriti\ or\ OPG + FIC\ antibiotic) \qquad \text{Eq (7)}$$

## Statistical analysis

Statistical analysis was performed using Graph Pad Prism version 5.0 software. Results were expressed as mean and standard deviation. First, the data obtained for the analyzes related to antioxidant potential and antimicrobial activity and modulation of antibiotic action were evaluated for normality using the Shapiro-Wilk test. Data were evaluated using the Student's t-test to determine the antioxidant potential through CAT and ABTS tests and assess antimicrobial activity and cytotoxicity. An analysis of variance and Tukey's post-test were used for the reducing power. A value of $p < 0.05$ was considered statistically significant in all cases.

## Results and discussion

### Characterization of porcine gelatin-based nanoparticles containing buriti oil (OPG)

The micrographs obtained through Scanning Electron Microscopy (SEM) analysis (Fig 1A) showed particles with a spherical shape, smooth surface without cracks and physical size at the nanometer scale (<100 nm), demonstrating good protection of the core through the encapsulation. The results obtained by Castro et al. [20] for buriti oil nanoencapsulated in porcine gelatin (OPG) were similar. Lira et al. [22] showed the same characteristics observed for shape and surface in all formulations containing quinoa oil.

The data obtained for particle diameter using DLS (Fig 1B) showed mean diameter and polydispersion index for OPG equal 72.0(0.68) and 0.371(0.0251), respectively. Thus, it is possible to state that OPG presented unimodal size distribution and reinforced the physical diameter observed by SEM, confirming that it is a nanoparticle. The nanometric particle size enables the food industry to improve the bioavailability of poorly soluble substances, such as lipids and natural antioxidants [34].

Castro et al. [20] found values of 51.00 (6.07) nm and 0.40 (0.05) for particle size and polydispersion index for OPG, respectively, constituting values close to those found in this work. Particles with a large diameter and size distribution can affect the texture and compromise the incorporated bioactive compounds, which did not occur in this study. The polydispersion index of less than 1 indicates homogeneity in the analyzed particle [20,35]. Thus, a particle that presents itself more homogeneously is inserted in a product more efficiently, not affecting the texture, therefore being positive for consumer use [35,36].

Fig 1C shows the FTIR spectra obtained for nanoencapsulated buriti oil, buriti oil, Tween 20, porcine gelatin. Hydrocarbon groups were detected in the buriti oil spectrum (Fig 1C, line b) by vibrational bands in the range of 2924–2852 cm$^{-1}$ (C-H), in addition to bands in the region of 3007 cm$^{-1}$ and 1462 cm$^{-1}$, which respectively characterize the–OH and–CH$_3$ groups, meaning that they reflect the strong constitution of this oil described in the literature as a source ingredient of carotenoids [16]. The stretching of the vibrational band at 1743 cm$^{-1}$

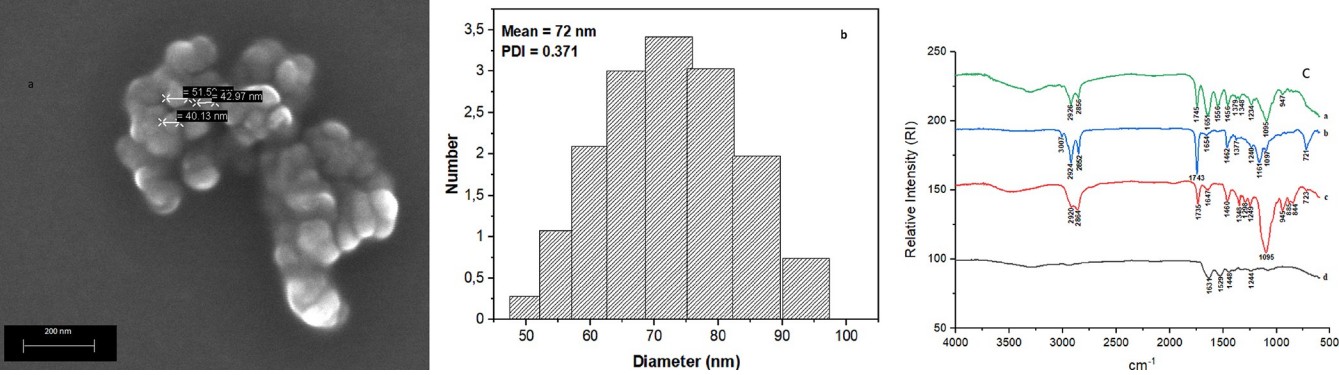

**Fig 1. Characterization of buriti oil nanoencapsulated in porcine gelatin.** A. Micrograph obtained by Scanning Electron Microscopy. B. Dynamic light scattering. C. Fourier Transform Infrared Spectroscopy of nanoencapsulated buriti oil (a), buriti oil (b), Tween 20 (c), porcine gelatin (d).

indicates the presence of double bonds in the oil chain (C = C; C = O), which shows the content of unsaturated fatty acids [37].

The vibrational band at 1651 cm$^{-1}$ in the porcine gelatin spectrum (Fig 1C, line d) reflects the presence of the C = O bond (amide I), according to Silverstein & Webster [37]. Another band is observed at 1529 cm-1, indicating amide (N-H) [38].

The spectrum of the Tween 20 surfactant (Fig 1C, line c) presented vibrations at 2920 cm$^{-1}$ and 2864 cm$^{-1}$ referring to stretching of the asymmetric and symmetrical methylene vibrations; at 1735 cm$^{-1}$, referring to the C = O connection; and at 1095 cm$^{-1}$, referring to the vibration stretching of–$CH_2$-O-$CH_2$-[37].

When observing the spectrum obtained for OPG (Fig 1C, line a), it can be seen that there was an interaction between porcine gelatin, buriti oil, and Tween 20 surfactant due to the attenuation and/or displacement of the vibrational bands that characterize the presence of the buriti oil (3007, 2924, 2852 and 1743 cm$^{-1}$), indicating protection of the oil in the obtained particles. The formation of new bands was also observed, which were not observed in the spectra of the raw materials (1556, 1348, and 1097 cm$^{-1}$), which may indicate hydrophobic interactions between buriti oil and the non-polar amino acids of porcine gelatin. Castro et al. [20] also observed chemical interactions between OPG constituents, meaning buriti oil with porcine gelatin.

The encapsulation efficiency determines the oil content (%) that was successfully encapsulated inside the particles, also being an indicator of free oil present on the surface of the particles [39]. The nanoformulation yield was 13.65g (1.14), and the result of the encapsulation efficiency was 89.56% (1.14). According to the literature is considered high efficiency, or when ≥ 80% [40,41].

Thus, this analysis confirmed the results obtained through the FTIR, which indicated a chemical interaction between gelatin, buriti oil, and Tween 20, which managed to provide greater oil retention. This result can be explained by the fact that proteins have physicochemical properties which favor forming and stabilizing emulsions than carbohydrates, thus allowing greater oil retention in the particles [20,42].

It was necessary to ensure that the new batch of synthesized nanoformulation had the same physical and chemical characteristics observed in Castro. et al. (2020). Therefore, it was necessary to carry out the characterization to confirm that the diameter, morphology, chemical interactions, and incorporation efficiency were similar to the batch obtained by Castro et al. (2020), ensuring that the material's functionality would be preserved and possible to continue the study. This step is of fundamental importance to guarantee the standardization of each batch of nanoformulation obtained.

Because of these characterization results, it can be stated that the process for obtaining nanoparticles used in this study is standardized and reproducible, which is essential to maintain the OPG nanoformulation properties even when obtaining new batches. Thus, it is possible to affirm that the nanoparticles developed have the same physical and chemical characteristics as those obtained by Castro et al. [20] as desired, being possible to proceed with the study for further investigation of physical and chemical characterization, safety, antioxidant potential, and antimicrobial activity.

## Zeta potential

Fig 2 shows OPG Zeta Potential values with the pH variation that resulted in a change in charge density around the nanoparticle. Low zeta potential values show a tendency to settle in an aqueous medium. Still, this low charge indicates steric stability of OPG as they are values with charges close to zero, so they can aggregate more quickly, but they also have eased to disperse [43].

As the nanoparticle is composed of porcine gelatin and has an amphoteric character, changes in pH alter the OPG surface charge between positive and negative values [21]. It is noteworthy that this change in surface charge influenced by the pH of the medium is an interesting feature when working with the application of nanoparticles in biofilms [44,45].

**Thermogravimetry (TG) and differential thermal analysis (DTA).**    The TG/DTA curves of crude buriti oil, OPG, porcine gelatin, and Tween 20 are shown in Fig 3.

Buriti oil presents TG mass loss onset at approximately 365˚C associated with three endothermic events in the DTA between 239 and 446˚C, which may be related to the complete

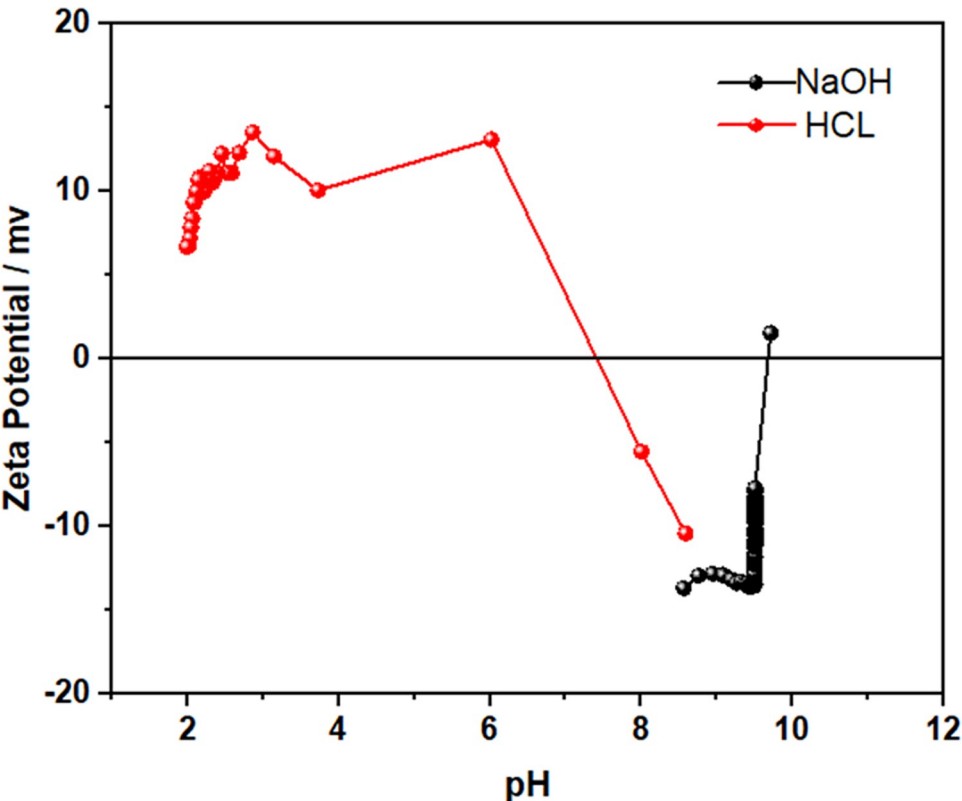

**Fig 2. Zeta Potential of buriti oil nanoencapsulated in porcine gelatin under different pH conditions.**

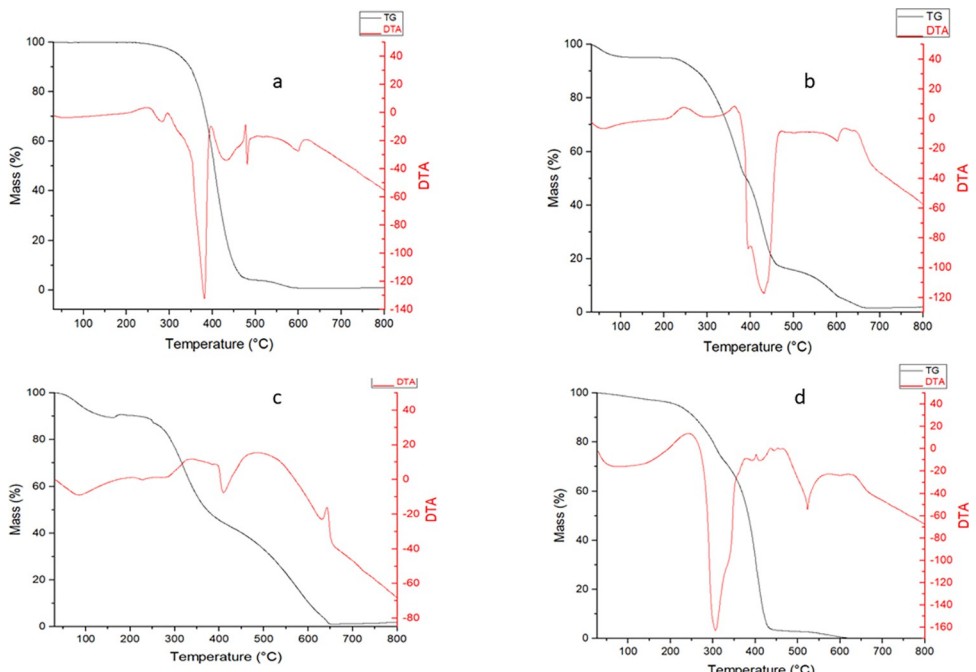

**Fig 3. Thermogravimetry graphs and differential thermal analysis.** a) Porcine gelatin b) Tween 20 c) Buriti oil d) OPG.

breakdown of fatty acids [46]. The total degradation of buriti oil ends at 596˚C and thus presents two stages of mass loss, with a total loss of 98.87%. The thermal stability of buriti oil is high when compared to other vegetable oils such as coconut oil (257˚C) and sesame oil (282˚C) [46,47].

The OPG TG curve in Fig 3B shows an initial and progressive mass loss up to approximately 100˚C, corresponding to the endothermic peak in the DTA, which may indicate water evaporation in the sample [48]. Then, the decomposition of the nanoparticle starts before the buriti oil at a temperature onset of 248˚C. After this event, mass loss occurs during the endothermic decomposition peak in both free buriti oil and nanoformulation. The mass loss is 95% and 83%, respectively, which may be related to the decomposition of the material wall [49]. The OPG DTA curve presents an exothermic peak characteristic of the sample crystallization, as the XRD indicates that the sample is partially amorphous, confirming this event.

The TG curve of porcine gelatin in Fig 3C had an initial thermal degradation between 35 and 160˚C, which may be related to the loss of adsorbed and bound water commonly seen with air-dried gelatin [45], confirmed by the endothermic peak in the DTA in the same temperature range. It still has the beginning of its degradation between 250˚C and 656˚C, with two stages of very evident mass loss and with a total loss of 98%. In the DTA curve, it is possible to confirm the degradation with two endothermic peaks and one exothermic peak in this same interval.

The Tween 20 TG curve in Fig 3D showed four decomposition stages. The first stage corresponds to water loss (32-196˚C) and the DTA curve presenting an endothermic peak in this temperature range, which corresponds to dehydration or desolvation. The beginning of Tween 20 degradation can be seen in the TG curve at 198˚C with the end of degradation at 654˚C, with total mass loss in the TG curve of 100%.

In evaluating the TG curves, it was possible to note that buriti oil is more stable than the other samples. However, although OPG has a lower initial degradation temperature than pure

**Table 1. The stages (TG) and mass loss of buriti oil, porcine gelatin, Tween 20 and OPG.**

| Sample | Number of decomposition steps | Start decomposition temperature/ ˚C | End decomposition temperature /˚C | Mass loss (%) |
|---|---|---|---|---|
| Buriti oil | 1 | 239.09 | 446.21 | 87.45 |
| | 2 | 446.00 | 596.00 | 11.42 |
| Porcine gelatin | 1 | 38.10 | 166.71 | 10.07 |
| | 2 | 252.69 | 380.57 | 38.66 |
| | 3 | 380.57 | 648.62 | 47.20 |
| Tween 20 | 1 | 32.84 | 196.17 | 3.80 |
| | 2 | 198.34 | 313.14 | 20.35 |
| | 3 | 313.14 | 432.29 | 71.42 |
| | 4 | 432.29 | 654.29 | 4.70 |
| OPG | 1 | 36.62 | 107.08 | 4.35 |
| | 2 | 218.00 | 388.47 | 44.32 |
| | 3 | 388.47 | 463.93 | 32.93 |
| | 4 | 463.93 | 672.59 | 15.93 |

buriti oil, it is still a very high temperature for a product with a greater possibility of handling and, consequently, application than pure oil. Tween 20 and porcine gelatin may have contributed to maintaining buriti oil's stability in the form of a nanoparticle and the same amount of degradation steps. Escobar-García et al. [50] performed a thermal analysis of oil rich in free eicosapentaenoic acid (EPA) and microencapsulated in concentrated whey protein and observed that the encapsulated sample had a delay in the degradation curve than the unencapsulated sample, with this result being attributed to the protection of the encapsulating agent [51].

Table 1 shows the steps of oil decomposition, encapsulating agent, Tween 20, and nanoencapsulated. Buriti oil has two degradation stages, with the first stage between 239.09˚C to 446.21˚C with a significant mass loss of 87.45%. The initial degradation temperature is similar to the data found for buriti oil in the study of Lima et al. (231.9˚C), but it differs concerning the mass loss, which was 59.6%. The results of the studies are often contradictory, as vegetable oils are very complex because they have a complex chemical composition, including lipid oxidation products, free fatty acids, phenolic compounds, glycerides, and non-glycerides that are present in oils in different proportions [52].

The physicochemical characteristics of buriti oils are different due to their location and other factors [53]. There are disagreements among authors regarding these degradation steps. Garcia et al. [51] found only two steps for the degradation of buriti oil since the mass loss was initially high. There are some hypotheses for these differences in vegetable oils, mainly regarding their fatty acid composition. Lima et al. [54] associated the mass losses in two stages: the triglycerides' evaporation and/or pyrolysis. Garcia et al. [51] reported the first and second weight loss as the oxidation of unsaturated and saturated fatty acids, respectively. The third step was reported as the decomposition of the polymer formed during the oxidation process.

OPG showed four degradation stages, and in the first stage, there was a mass loss of 4.35% at an initial temperature of 36.62˚C to 107.08˚C. Porcine gelatin and Tween 20 present an initial degradation temperature between 38.10˚C and 32.84˚C, respectively. This data suggests that the initial degradation observed in OPG may come from the degradation of gelatin and Tween 20. The second stage of nanoformulation degradation occurs at 218 to 388.47˚C, with a mass loss of less than 50%, partially corresponding to buriti oil. This data shows the OPG mass loss anticipation compared to free buriti oil, and however, the oil has a loss greater than 87%.

According to the data, it is observed that the stability of the free oil was higher in terms of initial decomposition temperature. Still, its degradation is very drastic, while the

**Table 2. The stages DTA of buriti oil, porcine gelatin, Tween 20, and OPG.**

| Sample | Stage | $T_{onset}$/ ˚C | $T_{peak}$/ ˚C | ΔH (J/g) |
|---|---|---|---|---|
| Buriti oil | 1 | 259.03 ↓ | 282.98 | 115.86 |
| | 2 | 350.09 ↓ | 381.57 | 2.85x10^6* |
| | 3 | 400.41 ↓ | 432.56 | 964.53 |
| | 4 | 476.98 ↓ | 480.94 | 98.64 |
| Porcine Gelatin | 1 | 37.69 ↓ | 84.81 | 509.39 |
| | 2 | 215.42 ↓ | 227.48 | 12.87 |
| | 3 | 399.46 ↓ | 409.59 | 370.38 |
| | 4 | 633.63 ↑ | 641.97 | 114.80 |
| Tween 20 | 1 | 31.77 ↓ | 78.14 | 592.78 |
| | 2 | 281.13 ↓ | 305.44 | 1.90x10^6* |
| | 3 | 397.64 ↑ | 402.05 | 6.03 |
| | 4 | 500.21↓ | 523.07 | 354.57 |
| OPG | 1 | 31.78 ↓ | 58.95 | 219.36 |
| | 2 | 214.05 ↓ | 245.47 | 286.40 |
| | 3 | 387.36 ↓ | 431.87 | 6.18x10^6* |
| | 4 | 583.34 ↓ | 601.43 | 154.45 |

*J/Kg; ↓ Endothermic ↑ Exothermic.

nanoformulation loses mass more slowly. It is important to emphasize that OPG is more stable than porcine gelatin and Tween 2 0 and that it has stability approaching that of pure buriti oil. In a way, OPG has significant advantages as it is a product with better handling, is more innovative, and maintains high oil stability. These data are important for the food area since nanoencapsulation exerts the possibility of using the product at different temperatures.

Table 2 presents the data of the DTA curves for buriti oil, encapsulating agent, Tween 20, and nanoencapsulated. It is possible to observe that all samples showed four endothermic or exothermic events characteristic of the analyzed samples. DTA can accurately measure the enthalpic transition temperatures of energy absorption and release. In this case, it is observed that the buriti oil and the nanoencapsulated present four characteristic endothermic events of their decomposition. The first endothermic decomposition event in buriti oil occurs at 259.03˚C with a maximum peak of 282.98˚C (ΔH = 115.86). In comparison, this event seems to happen at a lower temperature of 214.05˚C in OPG, with a maximum peak at 245.47˚C (ΔH = 286.40), corroborating the data obtained in the TG. It is also possible to visualize the endothermic peak in the nanoformulation characteristic of the dehydration of the encapsulating agent and Tween 20 around 31 and 78˚C.

## Cytotoxicity—MTT assay

The toxicity investigation of nanoparticles from essential oils, vegetables, and algae is necessary due to the unpredictable behavior that nanoscale materials have demonstrated [55]. The International Organization for Standardization (ISO 10993) compiles several tests on biomaterial/biological tissue interaction and determines that biocompatibility starts with evaluating material from cytotoxicity [56]. Part 5 of ISO 10993 is responsible for suggesting the tests and conditions necessary for evaluating the cytotoxicity test [56]. According to the standard, a reduction greater than 30% in cell viability in the tests is indicative of the material's cytotoxicity.

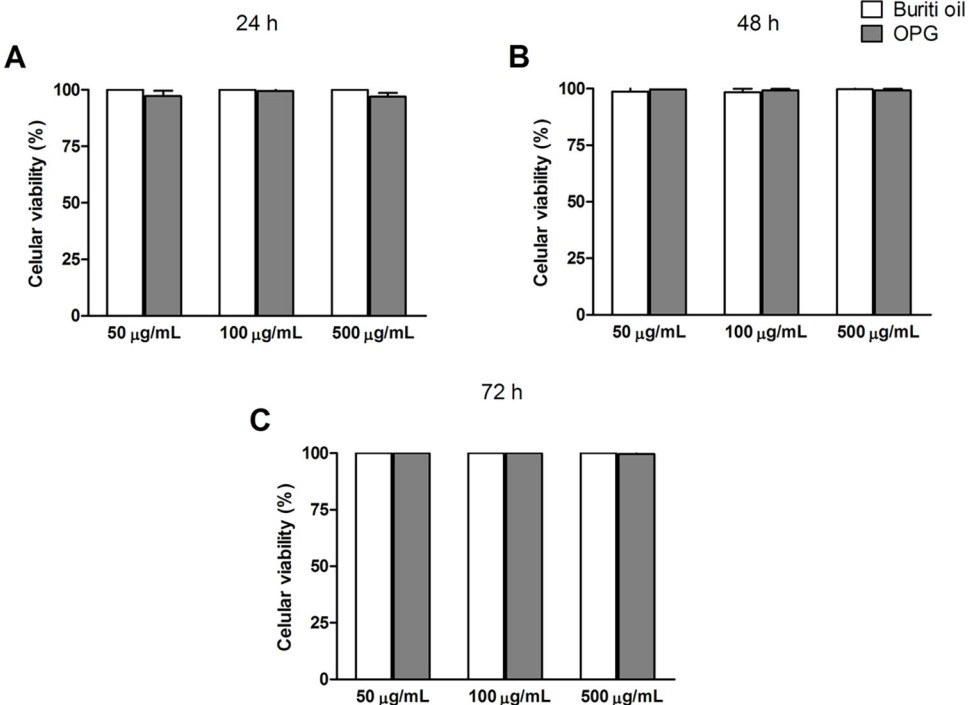

**Fig 4.** Cell viability (%) through the MTT assay in CHO-KI cells evaluated at different times (A– 24h, B– 48h, and C– 72h) and concentrations of buriti oil and OPG.

Fig 4 shows the MTT assay result. According to the data at the tested concentrations (0.5, 2.5, and 5 $\mu g.mL^{-1}$), crude buriti oil and OPG were not cytotoxic to CHO cells. There is no significant difference between buriti oil and OPG. Zanatta et al. [57] evaluated the cytotoxic effect of an emulsion containing buriti oil on 3T3 cells (1000 $mg.mL^{-1}$) and found no toxic effect, corroborating the results found in this work.

When reduced to < 70%, the viability has cytotoxic potential, which was not seen in this study [56]. Therefore, the buriti oil and the nanoformulation results are favorable because it does not present toxicity to the cells, assuming that their use is safe.

## Total antioxidant capacity (TAC)

TAC is determined using the phosphomolybdenum blue complex to reduce $Mo^{+6}$ to $Mo^{+5}$ by antioxidant compounds and the formation of green $Mo^{+5}$ complexes [58]. The results obtained (Table 3) showed that the nanoencapsulation process increased the antioxidant capacity of buriti oil threefold. This result can be explained by the greater dispersibility in water observed for the

**Table 3. Total antioxidant capacity of buriti oil and OPG.**

| Samples | mg AA. $g^{-1}$ |
|---|---|
| Buriti oil | 14.60 (1.63)[a] |
| OPG | 48.34 (3.71)[b] |

OPG: Buriti oil nanoencapsulated in a porcine gelatin.

AA: Ascorbic acid.

Mean and standard deviation (SD), n = 3. The different lowercase letters indicate a statistical difference.

OPG due to the obtained particle size, suggesting an increase in the contact surface for chemical interaction with water. Besides, chemical interactions between the materials of the system, as evidenced in the FTIR, consequently increased the antioxidant activity of the encapsulate [20,22].

TAC analyses for crude and encapsulated buriti oil were not found in the literature. Ribeiro et al. [59] evaluated the total antioxidant capacity of oil from faveleira seeds. The value found was 0.04 (0.00) mg AA. $g^{-1}$, constituting a lower result than its methanolic fraction [0.08 (0.00) mg AA. $g^{-1}$] and higher than its non-polar fraction [0.02 (0.00) mg AA. $g^{-1}$].

Silva et al. [60] extracted astaxanthin from shrimp of the *Litopenaeus vannamei* species using soybean oil and evaluated the antioxidant potential. The TAC results showed that the astaxanthin-pigmented soybean oil had higher antioxidant activity than soybean oil (control).

Zhong and Shahidi [61] compared the antioxidant activities of polar and non-polar compounds. They observed that a higher concentration of non-polar samples is needed to achieve the optimal antioxidant activity. These data corroborate this study since the buriti oil concentration required to present antioxidant activity was higher than that of OPG, which has a solubility in water.

## Reducing power

The reducing power assesses the ability of a sample to donate electrons in the presence of ferric chloride under acidic conditions and thus reduce $Fe^{+3}$ to $Fe^{+2}$ [62]. The results (Fig 5) showed that a lower OPG concentration than crude buriti oil was necessary to present similar antioxidant activity. Therefore, it is noteworthy that there is 0.98 mg of buriti oil in 2 mg.$mL^{-1}$ of OPG according to the encapsulation efficiency and considering the amount and proportion of the encapsulating agents. It is possible to note that this concentration was sufficient to promote a reduction from $Fe^{+3}$ to $Fe^{+2}$ in 100%.

On the other hand, 1 mg. $mL^{-1}$ of crude buriti oil promoted a reduction from $Fe^{+3}$ to $Fe^{+2}$ by only 0.74%. The encapsulate concentration must take into account that there are also encapsulating agents in addition to the oil (swine gelatin and Tween 20 in a ratio of 1:2.15). In addition, encapsulation efficiency must also be considered, as not all of the oil placed in the system was encapsulated.

Based on this, the result obtained demonstrates that the nanoencapsulation process enhanced the reducing power of buriti oil, with the reduction being greater as the concentration used in the test increases. No reduction power studies were found for crude or encapsulated buriti oil. In analyzing turmeric essential oil, Mau et al. [63] observed that the reducing power increases as the concentration of oil increases, corroborating the data found in this work.

## Antioxidant activity by 2,2'-azinobis-3-ethylbenzothiazoline-6-sulfonic acid (ABTS•)

Nanoencapsulation is known to increase the antioxidant potential of natural molecules [64–66]. ABTS radical scavenging was evaluated using buriti oil and OPG (Table 4).

The results obtained demonstrate that the technological process of nanoencapsulation favored the antioxidant activity since the amount of OPG used was about four times smaller than that of buriti oil to reach the IC50.

Lira et al. [22] performed an analysis for crude quinoa oil and its three nanoencapsulated formulations (encapsulated with porcine gelatin, encapsulated with whey protein, and porcine gelatin in aqueous phase 1, and encapsulated with whey protein in aqueous phase 1 and gelatin in aqueous phase 2). The IC 50 of the formulations was lower than that of the oil, corroborating the data found in this study.

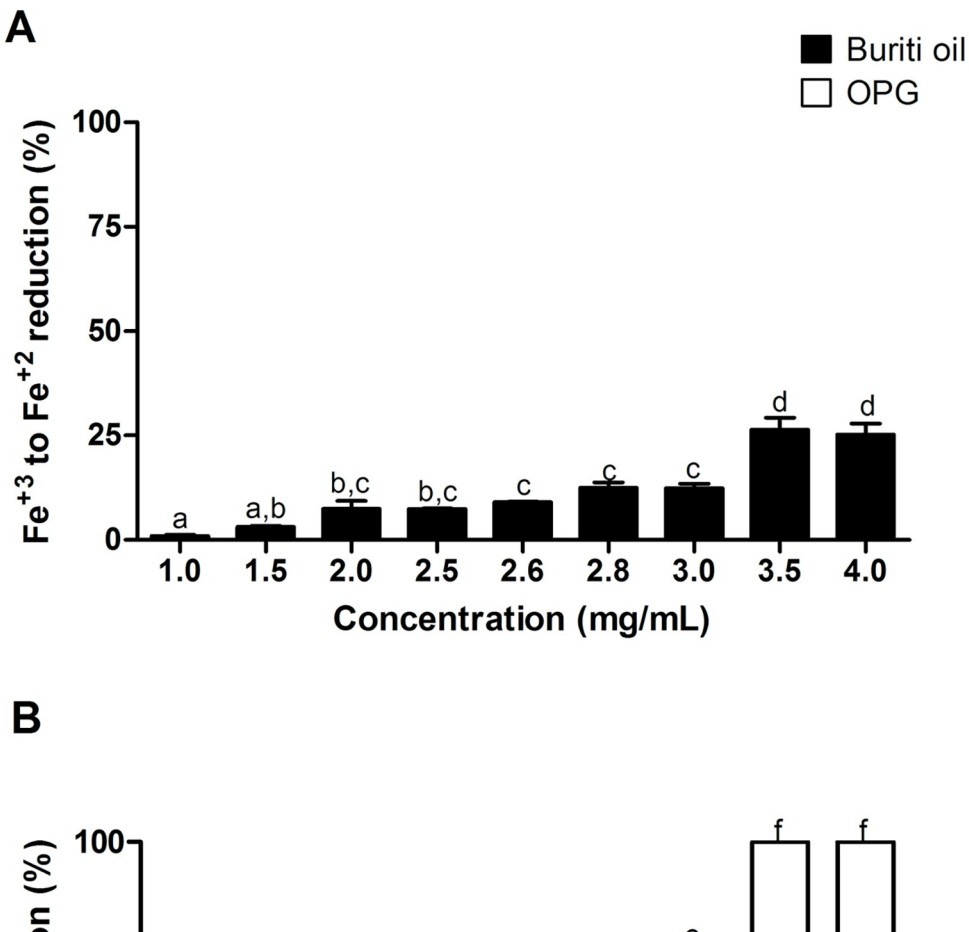

**Fig 5.** Reducing power test of crude buriti oil (A) and OPG nanoformulation. The data obtained presented parametric distribution. Therefore, the ANOVA test with Tukey's post-test was used to determine the significant differences. *Equal letters indicate that the values do not differ statistically ($p > 0.05$).

Eugenol essential oil (EO) was nanoencapsulated by the O/W emulsification method with sodium caseinate, maltodextrin and Tween 80 [67]. The results showed that the nanoencapsulated eugenol EO had a lower IC50 for the ABTS radical assay than crude eugenol [67].

This analysis showed that the OPG nanoparticle has greater antioxidant activity than its unencapsulated form. Thus, nanoscale diameter allows for increased contact surface for

**Table 4. Antioxidant activity (IC 50) buriti oil and OPG.**

| Samples | IC50 (mg. mL$^{-1}$) |
|---------|----------------------|
| Buriti oil | 4.18 (0.05)[a] |
| OPG | 0.94 (0.02)[b] |

OPG: Buriti oil nanoencapsulated in a porcine gelatin.

Mean and standard deviation (SD), n = 3. The different lowercase letters indicate a statistical difference.

chemical interactions, increased water dispersibility, bioaccessibility, and bioavailability compared to micro or macro scale size (12). Therefore, nanoencapsulated oils are protected against oxidation of their bioactive constituents and have a higher solubility in water [20,22,68]. This difference is relevant for its applications in food matrices to prolong the biological action and provide health benefits [65,69].

## Determination of the modulating activity of antibiotics

Due to the emergence of multidrug-resistant strains to antibiotics, combined drugs are used against these resistance mechanisms [70]. Antimicrobial resistance, in general, alters antibiotic action through the following mechanisms: modification of the antimicrobial target (decreased drug affinity), decreased drug absorption, activation of efflux mechanisms to expel the harmful compound (overexpression efflux pumps), or global changes in critical metabolic pathways through the modulation of regulatory networks [71]. One example is plant extracts that exhibit synergistic activity against microorganisms, as they present a complex mixture of natural products and present a low risk in the development of bacterial resistance [72]. Some natural products such as plant extracts, essential oils [73], and fixed oils [74] have modified the activity of antibiotics. Fig 6 shows the result of the MIC of buriti oil and OPG combined with norfloxacillin and gentamycin antibiotics.

Gentamycin is a drug used to treat infections, especially by Gram-negative bacteria, such as *E. coli* and *P. aeruginosa*, and it can also be used in combination [75]. Its MIC for *E. coli* and *P. aeruginosa* is 1 μg.mL$^{-1}$ and 2 μg.mL$^{-1}$, respectively [76,77]. Norfloxacin is a broad-spectrum antibiotic used for Gram-positive and Gram-negative bacteria [78]; it has a MIC of 25 μg.mL$^{-1}$ for *E. coli*, 2 μg.mL$^{-1}$ for *P. aeruginosa*, and 128 μg.mL$^{-1}$ for *S. aureus* [79–81]. Fixed buriti oil has a MIC >1024 for several strains, including *E. coli*, *P. aeruginosa*, and *S. aureus* [82].

Modulation with vegetable oils promotes better action on bacteria when there is a synergism of this combination, reducing the amount or increasing the medication used [82]. The mechanisms of action between antibiotic and oil [83,84] may be related to the interaction with the lipid bilayer and cell membrane, affecting the respiratory chain and energy production by the bacteria [85], or even interference with bacterial enzymatic systems that may enhance the mechanism of action of antibiotics in combination with oils [86].

The combination of antibiotics with crude buriti oil (Fig 5A) and OPG (Fig 6B) showed greater efficiency in inhibiting the action of the *E. coli* bacteria when compared to isolated antibiotics. The combination of norfloxacillin and gentamycin antibiotics with the crude buriti oil reduced the MIC by 93.75% and 75%, respectively. Combining the same antibiotics with OPG reduced the MIC of norfloxacillin by 98% and gentamycin by 75% in *E. coli*. For gentamycin, the nanoencapsulated oil maintained the modulating activity of the crude oil in 75%.

Leão et al. [87] observed that a nanoemulsion containing buriti oil inhibited the microbial growth of *E. coli* by 61% at a concentration of 3.14 μg.mL$^{-1}$, demonstrating a bacteriostatic effect. With this, the researchers observed that the physicochemical characteristics of the nanoemulsion contribute to its biological activity. The particle size can directly affect the

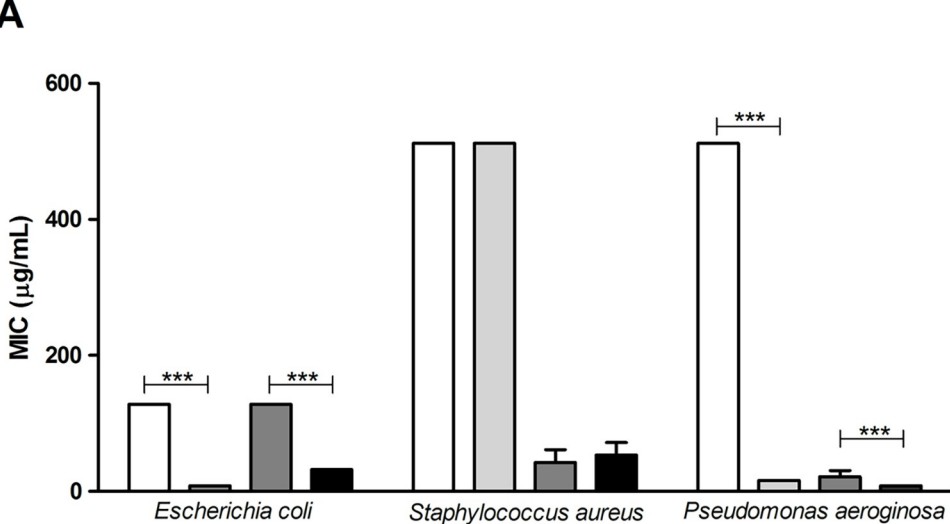

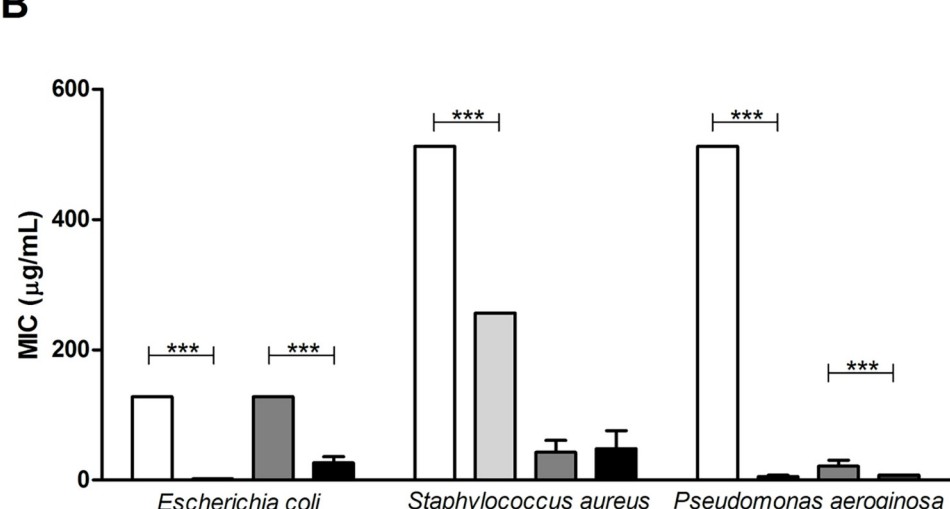

**Fig 6.** MIC values ($\mu$g.mL$^{-1}$) of antibiotics in a bacterial growth assay in the presence of buriti oil (A) and buriti oil nanoencapsulated in porcine gelatin (OPG) (B). The tests were performed under the following conditions: Norfloxacillin (white bar), norfloxacillin + crude buriti oil/OPG (light gray bar), gentamicin (dark gray bar), and gentamicin + crude buriti oil/OPG (black bar). ***p<0.0001.

antimicrobial activity of the emulsion based on essential oils, meaning a smaller size (nanometric scale) induces a greater inactivation of *E. coli* when compared to conventional emulsions [87].

Norfloxacillin alone, when compared with norfloxacillin combined with crude buriti oil (Fig 6A), did not present modulating activity for *S. aureus* with a statistically significant difference (p>0.05). However, the antibiotic activity combined with OPG decreased the MIC by 50% (Fig 6B). For gentamycin combined with crude oil such as OPG, there was an antagonism characterized by an increase in MIC by 100% in both demonstrated cases. Gentamycin does not act on Gram-positive bacteria, as *S. aureus* [75], showing that the nanoformulation maintained the crude buriti oil activity. Two main mechanisms are involved in antagonistic effects

**Table 5. Fractional inhibitory concentration (FIC) and fractional inhibitory concentration indices (FICi) of Buriti oil, OPG, norfloxacillin, and gentamycin.**

| Microorganism | MIC $_{buriti\ oil}$ ($\mu g.mL^{-1}$) | MIC $_{Norfloxacillin}$ ($\mu g.mL^{-1}$) | MIC $_{Norfloxacillin+Buriti\ oil}$ ($\mu g.mL^{-1}$) | FICi |
|---|---|---|---|---|
| *E. coli* | 64 | 128 | 8 | 0.2 |
| *P. aeruginosa* | 128 | 512 | 16 | 0.1 |
| | MIC $_{OPG}$ ($\mu g.mL^{-1}$) | MIC $_{Norfloxacillin}$ ($\mu g.mL^{-1}$) | MIC $_{Norfloxacillin+OPG}$ ($\mu g.mL^{-1}$) | FICi |
| *E. coli* | 16 | 128 | 2 | 0.1 |
| *P. aeruginosa* | 32 | 512 | 4 | 0.1 |

MIC: Minimum inhibitory concentration, FIC: Fractional inhibitory concentration, FICi: Fractional inhibitory concentration index, OPG: Buriti oil nanoencapsulated in porcine gelatin.

associated with natural products and antibiotics: chelation of antibiotic constituents by the natural product and competition of substances for the same binding site [88].

Norfloxacillin combined with buriti oil and OPG had greater inhibition for *P. aeruginosa* bacteria than norfloxacillin alone, with a statistical difference between them (p<0.05) and a 99% reduction in MIC with OPG and 97% with crude buriti oil. Gentamycin showed a significant difference in the two combinations (with free oil and OPG), in which there was a 50% reduction in MIC. In this case, both the free oil and OPG had a synergistic effect with the antibiotics.

Oils modify the antimicrobial effect depending on the antibiotic, oil tested and the bacterial species analyzed [14], which can be attributed to an interaction of the oil with the plasma membrane of the bacteria, increasing the permeability of the plasma membrane to the antibiotic [14,89,90].

Pereira et al. [82] used two antibiotics (gentamycin and amikacin) and two bacterial strains (*S. aureus* and *E. coli*) to verify the modulating activity of fixed buriti oil. The synergistic effect was observed in both antibiotics for *S. aureus*, with no MIC changes in *E. coli*.

The combination of buriti oil and OPG associated with norfloxacillin presenting FICi values ≤ 0.5 shows the synergistic effect for *E. coli* e *P. aeruginosa* (Table 5).

Combinations of natural products and antibiotics [91] can affect more than one target for bacterial growth inhibition. This strategy is known as "herbal shotgun" or "synergistic multi-effect targeting," in which different therapeutic components collaborate in a synergistic-agonistic way [88].

It is believed that both the antibacterial potential and the modulating activity of antibiotics attributed to vegetable oils are, at least in part, associated with the fatty acids present in the composition of the product since some fatty acids have already been shown to be able to improve the antimicrobial activity. And inhibit bacterial growth [15,92]. It is reported that the potential of vegetable oils to act as antibacterial modulators is in part associated with the detergent property of fatty acids against the amphipathic structure of bacterial cell membranes [15]. In this context, the synergistic effect observed against *E.coli* and *P. aeroginosa* may be associated with the detergent properties of fatty acids.

The literature points out that long-chain unsaturated fatty acids, such as oleic and palmitic, demonstrate antibacterial activity. In addition, the conjugated use of fatty acids and peptides or antibiotics can potentiate antimicrobial activity due to increased membrane permeability [93,94]. This ability to solubilize membrane components (lipids and proteins) creates gaps in this structure that will affect metabolic processes essential for acquiring energy for the bacterial cell, such as the electron transport chain and oxidative phosphorylation. Membrane damage can also inhibit enzyme activity and toxic peroxidation [95]. In addition, the presence of hydrophobic compounds in vegetable oils can increase the cell's permeability to antibiotics,

resulting in greater efficiency and reducing the minimum concentration necessary for the antibiotic to act against the bacteria [15,96].

The present study is innovative and unprecedented in the literature because it manages to evaluate the effect of the nanoformulation in the modulation of the antibiotic activity of known drugs, and not only of the non-encapsulated vegetable oil as presented in other studies. Thus, this study showed promising results for using free buriti oil and OPG in association with modifying antibiotic resistance against multiresistant strains such as *E. coli* and *P. aeroginosa*.

OPG is synthesized using low-cost materials. In addition, obtaining it is scalable, and drying by lyophilization is already used in the food industry. Thus, the nanoformulation has the potential to be incorporated into foods to add nutritional value due to the presence of buriti oil and its bioactive compounds, and also potential use in food packaging to act in the inhibition of pathogenic bacteria. Thus, OPG presents itself as a viable alternative to promote the preservation and/or enhancement of the oil's bioactive properties, which favors its use by the food and pharmaceutical industry, in addition to generating benefits to consumers' health. The application of nanotechnology in materials of plant origin promotes the valorization of this resource. It can contribute to the local development of the buriti-producing region within the perspective of circular economy and sustainable production. The present study shows, through FICi, the synergistic effect between the antibiotics gentamicin and norfloxacillin and the OPG against *E.coli* and *P. aeroginosa*.

## Supporting information

**S1 Dataset.**
(XLSX)

## Author Contributions

**Conceptualization:** Ana Paula Gomes Barreto.

**Formal analysis:** Neyna de Santos Morais, Gabriela Rocha Ramos.

**Investigation:** Neyna de Santos Morais, Victoria Azevedo Freire Ferreira.

**Methodology:** Susana Margarida Gomes Moreira, Gildácio Pereira Chaves Filho, Pedro Ivo Palacio Leite, Ray Silva de Almeida, Cícera Laura Roque Paulo, Rafael Fernandes, Sebastião Ânderson Dantas da Silva, Sara Sayonara da Cruz Nascimento, Francisco Canindé de Sousa Júnior.

**Project administration:** Cristiane Fernandes de Assis.

**Resources:** Gabriela Rocha Ramos, Victoria Azevedo Freire Ferreira, Gildácio Pereira Chaves Filho, Pedro Ivo Palacio Leite, Ray Silva de Almeida, Cícera Laura Roque Paulo, Rafael Fernandes, Sebastião Ânderson Dantas da Silva, Sara Sayonara da Cruz Nascimento, Francisco Canindé de Sousa Júnior.

**Supervision:** Thaís Souza Passos, Cristiane Fernandes de Assis.

**Writing – original draft:** Neyna de Santos Morais, Ana Paula Gomes Barreto.

**Writing – review & editing:** Thaís Souza Passos, Cristiane Fernandes de Assis.

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
