## [Decision Letter · Decision Letter 0]

10 Jan 2022

PONE-D-21-38955Nanoencapsulation of buriti oil in porcine gelatin enhances the antioxidant potential and improves the effect on the antibiotic activity modulationPLOS ONE

Dear Dr. de Assis,

Thank you for submitting your manuscript to PLOS ONE. After careful consideration, we feel that it has merit but does not fully meet PLOS ONE’s publication criteria as it currently stands. Therefore, we invite you to submit a revised version of the manuscript that addresses the points raised during the review process.

We look forward to receiving your revised manuscript.

Kind regards,

Thanh-Danh Nguyen, PhD

Academic Editor

PLOS ONE

Journal Requirements:

"This study was partly financed by the

Coordenação de Aperfeiçoamento de Pessoal de Nível Superior - Brasil (CAPES) - Finance Code 001."

"This study was partly financed by the

Coordenação de Aperfeiçoamento de Pessoal de Nível Superior - Brasil (CAPES) - Finance Code 001."

Reviewers' comments:

Reviewer's Responses to Questions

**Comments to the Author**

1. Is the manuscript technically sound, and do the data support the conclusions?

Reviewer #1: Partly

Reviewer #2: Partly

2. Has the statistical analysis been performed appropriately and rigorously? 

Reviewer #1: Yes

Reviewer #2: Yes

3. Have the authors made all data underlying the findings in their manuscript fully available?

Reviewer #1: Yes

Reviewer #2: Yes

4. Is the manuscript presented in an intelligible fashion and written in standard English?

Reviewer #1: Yes

Reviewer #2: Yes

5. Review Comments to the Author

Reviewer #1: The authors are carried out research for nanoformulation of buriti oil by using porcin gelatin in a scientific way, however, the need to describe the criteria was fixed in the selection of buriti fruits and formulation or production process is not well described.

In the introduction part, the research gap is not clear?

The results and discussion part have a good delivery to the information but lacking in scientific support studies.

This study was already carried out by Castro et al. and Garcia et al. in terms of Characterization OPG and what is your innovation or what would you tell to the scientific community or food industrialist?.

Units are not well described as per author guidelines.

Grammatical mistakes appeared in the manuscript and it was corrected the manuscript.

Mandatory for reframing the conclusion part in the first 4 lines, the present form of conclusion is not opt for the manuscript.

Figure quality is not good and mandatory need to improve the quality in all figures.

Corrections were mentioned in the manuscript and need to carry out.

Check the references and some of the references were missing in DOI?

Reviewer #2: The findings of the investigation are interesting, it uses suitable experiments to show the characteristic of the nanoencpasulated material and its action for antibiotic activity modulation. However there are some weakness in the manuscript that need to be observed. I suggest to accept the manuscript after major revision. My comments are given below.

1. How encapsulation facilitates the delivery of buriti oil. Author should perform the release experiment for its practical application.

2. Why nanoencapsulation increased the antioxidant activity of OPG

3. Why author selected porcine gelatin as wall polymer, state the prominent reason. There are other biocompatible polymers also.

4. Authors demonstrated herbal shotgun or synergistic multi-effect targeting in this investigation. This is very interesting finding. But, how author confirmed that the efficacy was due to synergism. In support of this statement statistical model showing the synergistic data is required.

5. SEM micrograph is not very clear.

6. Author found the combination of antibiotics with crude buriti oil and OPG showed greater efficiency in inhibiting the action of the E. coli bacteria when compared to isolated antibiotics. Why this enhancement in efficacy was found. Give the prominent reason.

7. Before adding commercial value to buriti oil, investigation about cost effectiveness is an important parameter to be considered. Author must incorporate the cost effectivity for encapsulation and how it can be commercially used?

8. Author presented this in line 142-145 as “After the encapsulation process, the resulting emulsion was subjected to drying by lyophilization (LioTop L101) at -57ºC and pressure of 43 μHg for further haracterization, evaluation of cytotoxicity, antioxidant potential, and effect on the modulation of antibiotic activity”. Why author performed the experiments of cytotoxicity, antioxidant potential, and antibiotic modulatory activity by lyophilized sample? What is the difference, if the experiment was performed through prepared emulsion? I think, after lyophilization, there is a change of aggregation and agglomeration of particles, therefore, chances of experimental hindrance about the results.

9. Why Tween 20 was used as surfactant in this investigation? Why not Tween-80, based on HLB values it is more used for nanoencapsulation.

6. PLOS authors have the option to publish the peer review history of their article (what does this mean?). If published, this will include your full peer review and any attached files.

Reviewer #1: **Yes: **THIRUKKUMAR SUBRAMANI

Reviewer #2: No

---

## [Author Response · Author response to Decision Letter 0]

22 Feb 2022

Dear Editor-in-Chief at the PLos One,

 Thank you for considering our manuscript entitled "Nanoencapsulation of buriti oil in porcine gelatin enhances the antioxidant potential and improves the effect on the antibiotic activity modulation" for revision. We have carefully read the reviewers' comments, answered all questions in this letter, and made the requested changes in the revised manuscript (highlighted in yellow). We hope the revised manuscript is now suitable for publication in PLos One. 

Best regards,

Cristiane Fernandes de Assis 

Department of Pharmacy 

Federal University of Rio Grande do Norte

Phone: (55 84) 99152-7007

E-mail: cristianeassis@hotmail.com

Answers to reviewer #1:

The authors are carried out research for nanoformulation of buriti oil by using porcin gelatin in a scientific way, however, the need to describe the criteria was fixed in the selection of buriti fruits and formulation or production process is not well described. In the introduction part, the research gap is not clear? 

Thank you for the comment. The suggestions were accepted. We made the proper changes in the results text, highlighted in yellow. The buriti oil used in the study was kindly donated by a company (Plantus® S/A) with an IBD certificate, which markets the product in Brazil and exports it to other countries. Regarding the nanoparticle synthesis process, the protocol followed was the same as that standardized by Castro et al. (1) (publication of our research group), aiming at obtaining a new batch of OPG with the same physical and chemical characteristics observed by Castro et al.(1) to continue the investigations related to the effect of nanoencapsulation on the functionality of buriti oil.

The results and discussion part have a good delivery to the information but lacking in scientific support studies. This study was already carried out by Castro et al. and Garcia et al. in terms of Characterization OPG and what is your innovation or what would you tell to the scientific community or food industrialist? 

Thank you for the comment. Our research group has been developing studies with nanoencapsulated vegetable oils (1,2). The present study is a continuation of the study of Castro et al. (1) that investigated the nanoencapsulation of buriti oil using a different encapsulating agent [porcine gelatin (OPG) and the combination of sodium alginate and porcine gelatin (OAG)], to evaluate the effect on water dispersibility and antimicrobial activity. The investigation pointed to best results for OPG, the smallest particle size [51.0 (6.07) nm], chemical interaction between the materials present in the system, greater dispersibility in water [85.62% (7.82)]. It potentiated the antimicrobial activity of buriti oil in 59%, 62%, and 43% against the bacteria Pseudomonas aeruginosa, Klebsiella pneumonia, and Staphylococcus aureus, respectively. 

Therefore, it is of fundamental importance to deepen the knowledge regarding the effects of nanoencapsulation on the biological properties of buriti oil to assess the antioxidant potential and modulating action on antibiotics of nanoparticles based on porcine gelatin containing buriti oil (OPG). 

Thus, it was necessary to ensure that the new batch of synthesized nanoformulation had the same physical and chemical characteristics observed in Castro. et al.(1) Therefore, it was necessary to carry out the characterization to confirm that the diameter, morphology, chemical interactions and incorporation efficiency were similar to the batch obtained by Castro et al. (1), ensuring that the material's functionality would be preserved and possible to continue the study. This step is of fundamental importance to guarantee the standardization of each batch of nanoformulation obtained.

The present study is innovative and unprecedented in the literature because it manages to evaluate the effect of the nanoformulation in the modulation of the antibiotic activity of known drugs, and not only of the non-encapsulated vegetable oil as presented in other studies. Interest in products developed through Nanotechnology has increased not only in the food industry but also in the pharmaceutical industry due to properties such as the high dissolution rate associated with the increased permeability of the active compound through the intestinal wall, increased physical and chemical stability of the core, the ease of handling, transport, and incorporation into the matrix of a product. In addition, to enabling water solubility, bioaccessibility and bioavailability of lipophilic substances, preservation and enhancement of functionalities, and enabling the reduction of the concentration of the bioactive of interest necessary to achieve the desired biological effect.

Units are not well described as per author guidelines. Grammatical mistakes appeared in the manuscript and it was corrected the manuscript. Mandatory for reframing the conclusion part in the first 4 lines, the present form of conclusion is not opt for the manuscript. Figure quality is not good and mandatory need to improve the quality in all figures. Corrections were mentioned in the manuscript and need to carry out. Check the references and some of the references were missing in DOI?

Thank you for the comment. The suggestions were accepted. We made the proper changes in the results text, highlighted in yellow. We have carefully addressed the manuscript's grammar, usage, and overall readability. This English review was carried out by a native English speaker with experience in international research. We also used the software Grammarly (full version) to further check for gross mistakes. We are fully available to respond to any further questions, attending to other suggestions. Besides, the units and conclusions were adjusted based on the author's guidelines. The reference list was revised for the insertion of the DOI, and the resolution of the figures was improved. 

Answers to reviewer #2: 

The findings of the investigation are interesting, it uses suitable experiments to show the characteristic of the nanoencpasulated material and its action for antibiotic activity modulation. However there are some weakness in the manuscript that need to be observed. I suggest to accept the manuscript after major revision. My comments are given below. 

Thank you for considering our manuscript “Nanoencapsulation of buriti oil in porcine gelatin enhances the antioxidant potential and improves the effect on the antibiotic activity modulation" for revision. 

1. How encapsulation facilitates the delivery of buriti oil. Author should perform the release experiment for its practical application. 

The present study is a continuation of a study previously published by our research group (Castro et al.,(1), which proposed to enable the nanoencapsulation of buriti oil, aiming to obtain a formulation with a nanoscale diameter, distribution of homogeneous size, capable of allowing the dispersibility of the oil in water, and potentiating the antimicrobial activity, that is, increasing the action using a smaller amount of buriti oil. Thus, Castro et al.(1) tested different materials (sodium alginate, porcine gelatin, the combination of both) and obtained a formulation (OPG – nanoencapsulated buriti oil in porcine gelatin) that achieved the expected results. Based on this, the main focus of the present study was to deepen the investigation of the effect of nanoencapsulation on the buriti oil functionalities, such as the antioxidant activity and modulation of the antimicrobial activity of known antibiotics and the impact on thermal stability. In addition, it was of fundamental importance to evaluate the toxicity of OPG in cells, aiming at future studies in animal models. Therefore, future studies by our research group will determine in vitro simulated digestion to obtain information on controlled release, antioxidant and anti-inflammatory effects, complementing the information with studies in eutrophic animal models and diet-induced obesity.

2. Why nanoencapsulation increased the antioxidant activity of OPG

Nanoscale diameter allows for increased contact surface for chemical interactions, increased water dispersibility, bioaccessibility, and bioavailability compared to micro or macro scale size (3). Therefore, nanoencapsulated oils are protected against oxidation of their bioactive constituents and have a higher solubility in water (1,2,4). These associated factors guarantee an increase in the antioxidant potential.

3. Why author selected porcine gelatin as wall polymer, state the prominent reason. There are other biocompatible polymers also. 

Porcine gelatin is a low-cost material, widely used in the food, cosmetic and pharmaceutical industry, as a coating, binding, gelling agent in confectionery, capsules and food supplements, creams, lotions, among others, due to its structural stability, and physicochemical, functional and nutritional properties. It can act as an emulsifier in oil-in-water emulsions, promoting an increase in the physical-chemical stability of polyunsaturated fats. In addition, the gelling property is of great interest for applications in the health area, as it helps in the adhesion of the particle to the mucosa to promote the controlled release.

4. Authors demonstrated herbal shotgun or synergistic multi-effect targeting in this investigation. This is very interesting finding. But, how author confirmed that the efficacy was due to synergism. In support of this statement statistical model showing the synergistic data is required. 

Thank you for the comment. The suggestions were accepted. We made the proper changes in the results text, highlighted in yellow. We inserted in the article the methodology and the result (Table 5) in which we demonstrate the synergistic effect of buriti oil / OPG and antibiotics.

5. SEM micrograph is not very clear. 

Thank you for the comment. The resolution of the micrographs was improved using a tool indicated by PLos One.

6. Author found the combination of antibiotics with crude buriti oil and OPG showed greater efficiency in inhibiting the action of the E. coli bacteria when compared to isolated antibiotics. Why this enhancement in efficacy was found. Give the prominent reason. 

According to published studies, the antibacterial potential and modulating activity of antibiotics attributed to oils are, at least in part, associated with the fatty acids present in their composition since some fatty acids have already been shown to be able to improve antibiotic activity and inhibit the growth of bacteria (5,6). Studies suggest that oils' potential to act as antibacterial or antibiotic modulators may be associated with the detergent property of fatty acids against the amphipathic structure of bacterial cell membranes (7). This detergent property can act on the membrane (lipids and proteins), creating pores in the structure that will affect the metabolic processes of bacteria that are essential for their survival (8). Furthermore, the hydrophobic compounds in oils can increase the cell's permeability to the antibiotic, increasing the efficiency and decreasing the MIC of the antibiotic (5,9). It should be noted that these studies and possible mechanisms of action were included in the discussion of the present study, and the changes were highlighted in yellow.

7. Before adding commercial value to buriti oil, investigation about cost effectiveness is an important parameter to be considered. Author must incorporate the cost effectivity for encapsulation and how it can be commercially used?

Thank you for the comment. The suggestions were accepted. We made the proper changes in the results text, highlighted in yellow. The nanoformulation is synthesized using low-cost materials. In addition, obtaining it is scalable, and drying by lyophilization is already used in the food industry. Thus, the nanoformulation has the potential to be incorporated into foods to add nutritional value due to the presence of buriti oil and its bioactive compounds, and also potential use in food packaging to act in the inhibition of pathogenic bacteria.

8. Author presented this in line 142-145 as “After the encapsulation process, the resulting emulsion was subjected to drying by lyophilization (LioTop L101) at -57ºC and pressure of 43 μHg for further characterization, evaluation of cytotoxicity, antioxidant potential, and effect on the modulation of antibiotic activity”. Why author performed the experiments of cytotoxicity, antioxidant potential, and antibiotic modulatory activity by lyophilized sample? What is the difference, if the experiment was performed through prepared emulsion? I think, after lyophilization, there is a change of aggregation and agglomeration of particles, therefore, chances of experimental hindrance about the results. 

Based on the literature (10,11) one of the advantages of the emulsification technique is to use the emulsions obtained in fluid or solid form. Therefore, powder forms can be obtained by drying the fluid emulsion by spray-drying or lyophilization. Thus, the drying of emulsions expands the range of applications in food, packaging, cosmetics, and pharmaceuticals, as it facilitates the dispersion of the material in an aqueous matrix. Based on this, the emulsion obtained was dried to expand its use in several products and not just to carry out the investigations presented in this study.

9. Why Tween 20 was used as surfactant in this investigation? Why not Tween-80, based on HLB values it is more used for nanoencapsulation.

Based on published studies, such as the one by Yuan et al. (12) , from the group of polysorbates, Tween 20 stands out as the surfactant with the highest hydrophilic-lipophilic balance (HLB), approximately 16.7, being possible to promote the stabilization of the interface between oil and water, allowing to obtain particles of smaller size compared to other polysorbates.

REFERENCES

1. Castro GMMA, Passos TS, Nascimento SS da C, Medeiros I, Araújo NK, Maciel BLL, et al. Gelatin nanoparticles enable water dispersibility and potentialize the antimicrobial activity of Buriti (Mauritia flexuosa) oil. BMC Biotechnol. 2020;20(1):1–13. 

2. Lira KHD da S, Passos TS, Ramalho HMM, Rodrigues D da SR, Vieira É de A, Cordeiro AMT de, et al. Whey protein isolate-gelatin nanoparticles enable the water-dispersibility and potentialize the antioxidant activity of quinoa oil (Chenopodium quinoa). PLoS One. 2020;30:1–17. 

3. Tabibiazar M, Davaran S, Hashemi M, Homayonirad A, Rasoulzadeh F, Hamishehkar H, et al. Design and fabrication of a food-grade albumin-stabilized nanoemulsion. Food Hydrocoll [Internet]. 2015;44:220–8. Available from: http://dx.doi.org/10.1016/j.foodhyd.2014.09.005

4. de Campo C, dos Santos PP, Costa TMH, Paese K, Guterres SS, Rios A de O, et al. Nanoencapsulation of chia seed oil with chia mucilage (Salvia hispanica L.) as wall material: Characterization and stability evaluation. Food Chem [Internet]. 2017;234:1–9. Available from: http://dx.doi.org/10.1016/j.foodchem.2017.04.153

5. Saraiva RA, Matias EFF, Coutinho HDM, Costa JGM, Souza HHF, Fernandes CN, et al. Synergistic action between Caryocar coriaceum Wittm. fixed oil with aminoglycosides in vitro. Eur J Lipid Sci Technol. 2011;113(8):967–72. 

6. Pereira YF, Costa MDS, Tintino SR, Rocha JE, Rodrigues FFG, Feitosa MK de SB, et al. Modulation of the antibiotic activity by the Mauritia flexuosa (Buriti) fixed oil against methicillin-resistant staphylococcus aureus (MRSA) and other multidrug-resistant (MDR) bacterial strains. Pathogens. 2018;7(4):1–8. 

7. Chu-Kung AF, Bozzelli KN, Lockwood NA, Haseman JR, Mayo KH, Tirrell M V. Promotion of peptide antimicrobial activity by fatty acid conjugation. Bioconjug Chem. 2004;15(3):530–5. 

8. Desbois AP, Smith VJ. Antibacterial free fatty acids: activities, mechanisms of action and biotechnological potential. Appl Microbiol Biotechnol [Internet]. 2010 Jan 15;85:1629+. Available from: https://link.gale.com/apps/doc/A229606459/AONE?u=capes&sid=bookmark-AONE&xid=82eb837a

9. Sousa EO de, Rodrigues FFG, Campos AR, Costa JGM da. <b>Phytochemical analysis and modulation in aminoglycosides antibiotics activity by Lantana camara L. Acta Sci Biol Sci [Internet]. 2015 Aug 6;37(2):213. Available from: http://periodicos.uem.br/ojs/index.php/ActaSciBiolSci/article/view/22877

10. Ferreira CD, Nunes IL. Oil nanoencapsulation: development, application, and incorporation into the food market. Nanoscale Res Lett. 2019;14. 

11. Comunian TA, Silva MP, Moraes ICF, Favaro-Trindade CS. Reducing carotenoid loss during storage by co-encapsulation of pequi and buriti oils in oil-in-water emulsions followed by freeze-drying: Use of heated and unheated whey protein isolates as emulsifiers. Food Res Int [Internet]. 2020;130:108901. Available from: http://www.sciencedirect.com/science/article/pii/S0963996919307872

12. Yuan Y, Gao Y, Zhao J, Mao L. Characterization and stability evaluation of β-carotene nanoemulsions prepared by high pressure homogenization under various emulsifying conditions. Food Res Int [Internet]. 2008;41(1):61–8. Available from: http://www.sciencedirect.com/science/article/pii/S0963996907001627

---

## [Decision Letter · Decision Letter 1]

7 Mar 2022

Nanoencapsulation of buriti oil (Mauritia flexuosa L.f.) in porcine gelatin enhances the antioxidant potential and improves the effect on the antibiotic activity modulation

PONE-D-21-38955R1

Dear Dr. de Assis,

We’re pleased to inform you that your manuscript has been judged scientifically suitable for publication and will be formally accepted for publication once it meets all outstanding technical requirements.

Kind regards,

Thanh-Danh Nguyen, PhD

Academic Editor

PLOS ONE

Additional Editor Comments (optional):

Reviewers' comments:

Reviewer's Responses to Questions

**Comments to the Author**

1. If the authors have adequately addressed your comments raised in a previous round of review and you feel that this manuscript is now acceptable for publication, you may indicate that here to bypass the “Comments to the Author” section, enter your conflict of interest statement in the “Confidential to Editor” section, and submit your "Accept" recommendation.

Reviewer #1: All comments have been addressed

Reviewer #2: All comments have been addressed

2. Is the manuscript technically sound, and do the data support the conclusions?

Reviewer #1: Yes

Reviewer #2: Yes

3. Has the statistical analysis been performed appropriately and rigorously? 

Reviewer #1: Yes

Reviewer #2: Yes

4. Have the authors made all data underlying the findings in their manuscript fully available?

Reviewer #1: Yes

Reviewer #2: Yes

5. Is the manuscript presented in an intelligible fashion and written in standard English?

Reviewer #1: Yes

Reviewer #2: Yes

6. Review Comments to the Author

Reviewer #1: This paper was scientifically well explained and comments to be well rectified.

However, I mentioned mistakes in paper itself. Carryout those mentioned mistakes and resubmit the paper to the journals also find the chances for minimizing the number of figures.

Reviewer #2: Please add one-two lines about FIC and synergistic activity at the last paragraph of the revised manuscript.

7. PLOS authors have the option to publish the peer review history of their article (what does this mean?). If published, this will include your full peer review and any attached files.

Reviewer #1: **Yes: **Thirukkumar Subrmani

Reviewer #2: **Yes: **Dr. Somenath Das

---

## [Editor Report · Acceptance letter]

11 Mar 2022

PONE-D-21-38955R1 

Nanoencapsulation of buriti oil (*Mauritia flexuosa L.f.*) in porcine gelatin enhances the antioxidant potential and improves the effect on the antibiotic activity modulation 

Dear Dr. de Assis:

I'm pleased to inform you that your manuscript has been deemed suitable for publication in PLOS ONE. Congratulations! Your manuscript is now with our production department. 

Kind regards, 

on behalf of

Dr. Thanh-Danh Nguyen 

Academic Editor

PLOS ONE